# Quantifying previous SARS-CoV-2 infection through mixture modelling of antibody levels

C. Bottomley [1,2 ✉], M. Otiende[2,3], S. Uyoga [3], K. Gallagher[2,3], E. W. Kagucia [3], A. O. Etyang[3], D. Mugo[3], J. Gitonga[3], H. Karanja[3], J. Nyagwange[3], I. M. O. Adetifa [2,3], A. Agweyu[3,4], D. J. Nokes [3,5], G. M. Warimwe[3,4] & J. A. G. Scott [2,3,4]

As countries decide on vaccination strategies and how to ease movement restrictions, estimating the proportion of the population previously infected with SARS-CoV-2 is important for predicting the future burden of COVID-19. This proportion is usually estimated from serosurvey data in two steps: first the proportion above a threshold antibody level is calculated, then the crude estimate is adjusted using external estimates of sensitivity and specificity. A drawback of this approach is that the PCR-confirmed cases used to estimate the sensitivity of the threshold may not be representative of cases in the wider population—e.g., they may be more recently infected and more severely symptomatic. Mixture modelling offers an alternative approach that does not require external data from PCR-confirmed cases. Here we illustrate the bias in the standard threshold-based approach by comparing both approaches using data from several Kenyan serosurveys. We show that the mixture model analysis produces estimates of previous infection that are often substantially higher than the standard threshold analysis.

[1] International Statistics and Epidemiology Group, London School of Hygiene & Tropical Medicine, London, UK. [2] Department of Infectious Disease Epidemiology, London School of Hygiene & Tropical Medicine, London, UK. [3] KEMRI-Wellcome Trust Research Programme, Kilifi, Kenya. [4] Nuffield Department of Medicine, Oxford University, Oxford, UK. [5] School of Life Sciences, University of Warwick, Coventry, UK.
✉email: christian.bottomley@lshtm.ac.uk

Establishing the amount of previous infection with SARS-CoV-2 is key to predicting the future impact of the virus. The evidence to date suggests that reinfection is uncommon, at least in the short term, and associated with mild disease[1,2]. Therefore, knowing how many people have previously been infected can help to establish to what extent a population is protected by natural infection.

The proportion previously infected is usually estimated from serological surveys (i.e. data on antibody levels). The conventional analysis of these data involves estimating the proportion above an arbitrary threshold and adjusting for the sensitivity and specificity at that threshold[3,4]. However, sensitivity is usually estimated using samples from PCR-positive cases who are symptomatic and have been recently infected. Since these samples typically have higher antibody levels than samples from the general population of previously infected individuals, including those with asymptomatic infection, this may lead to the overestimation of sensitivity and underestimation of the proportion previously infected[5]. Bias of this kind, that is bias that arises because sensitivity (or specificity) is estimated in a non-representative sample, is often referred to as spectrum bias. Mixture models offer an alternative approach to the analysis of serological data that does not involve specifying a threshold and is therefore not vulnerable to spectrum bias[6,7].

In this paper, we use data on antibody concentrations—optical density (OD) ratios measured by ELISA—from several Kenyan SARS-CoV-2 serosurveys to compare the standard threshold-based analysis with a mixture modelling approach.

In the mixture model, we assume the observed distribution of antibody levels is a mixture of two unobserved distributions—the distribution in individuals who have experienced previous infection and the distribution in those who have not. The model is therefore characterised by the two component distributions and the proportion in each component. For the uninfected component we specify that log antibody concentrations follow a normal distribution, and for the infected component we specify they follow a skew normal distribution[7]. To fit the model, we fix the variance of the uninfected component at a value estimated from pre-COVID-19 samples, and estimate the remaining parameters using a Markov chain Monte Carlo algorithm. We show that this mixture modelling approach generally produces higher estimates of the proportion previously infected than the standard threshold analysis.

## Results

We found that the positive (previously infected) and negative (previously uninfected) distributions estimated from the mixture model did not segregate as clearly as expected based on the distributions observed in pre-COVID-19 and PCR-positive samples (Figs. 1 and 2, Supplementary Table 1). In most surveys, this was because the positive distribution was shifted to the left relative to the distribution in PCR-positive samples, i.e. the mean was lower than in the PCR-positive samples (mean $\log_2$ OD ratios = 3.07). In contrast, the mean of the negative distribution was usually similar to the mean observed in the pre-COVID-19 samples (mean $\log_2$ OD ratios = −0.17). However, there was some variation by region, and in the surveys done in truck drivers the means were higher, while in the surveys done in pregnant women they were lower. In most surveys, the skew parameter of the positive distribution was close to zero, and the scale parameter, which determines the spread of the positive distribution, was similar to the standard deviation in PCR-confirmed cases (SD $\log_2$ OD ratios =1.32).

Because the mixture model predicted lower antibody levels in previously infected individuals than was observed in the

PCR-positive samples, the sensitivity of the threshold (i.e. the proportion of OD ratios >2 in the positive component) was generally lower than assumed in the standard threshold-based analysis; consequently the mixture model analysis produced higher estimates of the proportion previously infected than the threshold analysis (Fig. 3, Supplementary Fig. 1 and Supplementary Table 2). Across all surveys the mean sensitivity was 65% (cf. 93% sensitivity measured in the validation sample and assumed in the threshold analysis), and the estimated proportion previously infected was on average 1.44-fold higher than in the threshold analysis.

In general, the 95% credible intervals associated with the estimated proportions were wider in the mixture model analysis, with the largest differences occurring when there was strong overlap between the component distributions. For example, the three surveys of truck drivers (in Busia, Magarini and Malaba) produced the widest confidence intervals; they were also the surveys with the greatest overlap between the distributions.

To assess the robustness of our results, we fitted an alternative model where log antibody levels follow a two-component mixture distribution in previously infected individuals and a normal distribution in previously uninfected individuals, as in the original model. The two positive components do not have a clear biological interpretation, nevertheless we might imagine that recent/symptomatic infections predominate in the high-antibody-level component and older/asymptomatic infections predominate in the low-antibody-level component. The predicted distributions and estimates of the proportion positive from this alternative model were similar to those from the original skew normal model suggesting that the results are not sensitive to the distribution assumed for the positive component (Supplementary Fig. 2, Supplementary Fig. 3).

We further tested the mixture model in three simulated data scenarios (Supplementary Table 3). In scenario 1, the test data were generated by combining the PCR-positive and pre-COVID-19 samples. In scenarios 2 and 3 either the positive (scenario 2) or negative component (scenario 3) was simulated from a mixture of two normal distributions and the other component was simulated from a single normal distribution.

In keeping with the results of the previous sensitivity analysis, the mixture model performed well in scenarios 1 and 2: in both cases the estimated proportion positive was close to the expected value (scenario 1: 15% vs 14%; scenario 2: 19% vs 20%). By contrast, in scenario 3— where a mixture was used for the negative component—the mixture model estimate was severely biased (estimated = 36%, expected = 20%). This bias arises because the variance for the negative component is fixed to be equal to the variance estimated from pre-COVID-19 samples (this is one of the constraints used to fit the model). Consequently, variation that should be attributed to the negative component is instead attributed to the positive component thereby overestimating the proportion positive.

The findings from scenario 3 suggest that combining data across different populations may lead to the overestimation of previous infection. To test this hypothesis, we conducted a Kenya-wide analysis by combing the blood donor data and fitting the mixture model to these combined data. Our expectation was that the proportion previously infected would be overestimated in this analysis because of the variation in background antibody levels observed in the region-specific analyses (Supplementary Table 1). Specifically, we hypothesised that the estimate from the analysis of the combined data would be greater than average of the region-specific estimates, after weighting by the number of samples collected in each region. Consistent with this prediction, the estimate from the combined analysis was 44.3% whereas the average of region-specific analyses was only 18.9%. In fact, the

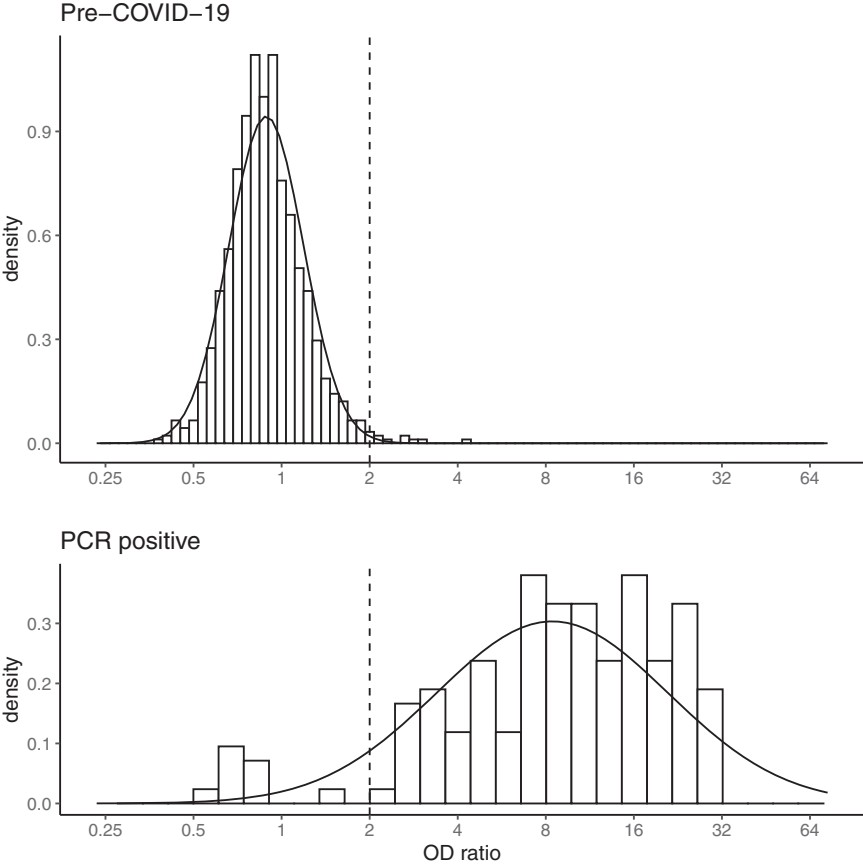

**Fig. 1 Distribution of anti-spike IgG antibodies in PCR-positive samples and pre-COVID-19 samples.** The dotted line indicates the threshold (OD ratio > 2) used to define seropositivity.

estimate from the combined analysis was greater than any of the region-specific estimates.

## Discussion

The mixture model analysis suggests that antibody levels are higher in samples from recent PCR-confirmed cases than in samples from previously infected serosurvey participants. Because of this, threshold-based estimates—which rely on having an accurate sensitivity estimate—underestimate previous infection.

There are two potential explanations for the higher antibody levels in recent PCR-confirmed cases. First, PCR-confirmed cases are more likely to be symptomatic, and symptoms such as cough, fever, and severe disease (hospitalisation) are positively associated with higher antibody levels[8–12]. Second, PCR testing is usually done soon after infection when antibody levels are high. In the PCR-positive samples used in our study, for example, the median time between symptom onset and blood sample collection was 21 days, which may have been ideal to capture the peak antibody response. By contrast, many survey participants will have been infected several months earlier and their antibody levels may have waned in the meantime. Antibody waning has been reported in a number of studies[8–11,13,14], and those that have specifically accounted for waning by assuming a constant rate of seroreversion—rather than by accounting for spectrum bias more generally as we have done—have predicted a significant impact on seroprevalence estimates[10,15,16]. In general, waning is greatest for anti-nucleocapsid antibodies, but it can also be significant for anti-spike protein antibodies. For example, in one study involving milder cases of infection the half-life of anti-spike antibodies was estimated to be just 73 days[17].

The mixture model results suggest variability in the background levels of anti-spike IgG between different populations. In addition to variation by region, we observed higher IgG levels in truck drivers and lower levels in pregnant women. This variation will bias the standard threshold analysis. For example, in a population with low background IgG levels, as observed in pregnant women, the specificity estimate will be too low, and the sensitivity estimate too high. The variation will also bias mixture model analyses if data are combined across different populations, as exemplified by the Kenya-wide analysis of blood donor data.

The reasons for the variation in baseline IgG levels are unclear. It could simply reflect temporal variation in the laboratory procedures, though the negative control should guard against this bias. Alternatively, it could be related to differences between populations in exposure to infection and possibly also infective dose. Several studies of pre-COVID-19 antibody levels have reported variation between populations, with antibody levels generally being higher in African populations than non-African populations[18,19]. Furthermore, anti-SARS-CoV-2 antibodies are known to cross-react with antibodies against other coronaviruses[20], and possibly also antibodies against dengue[21] and malaria[22], though the latter finding was not confirmed in a more recent study[23]. For pregnant women, it is possible that low antibody levels are a feature of immune environment in pregnancy[24].

The major limitation of the mixture modelling approach is that it is sensitive to the variance assumed for the uninfected population. Ideally the variance estimate should come from the population being surveyed, but in practice it will often be necessary to use an estimate from a different population. For example, we used pre-COVID-19 samples from blood donors in

**Blood Donors**

**Other surveys**

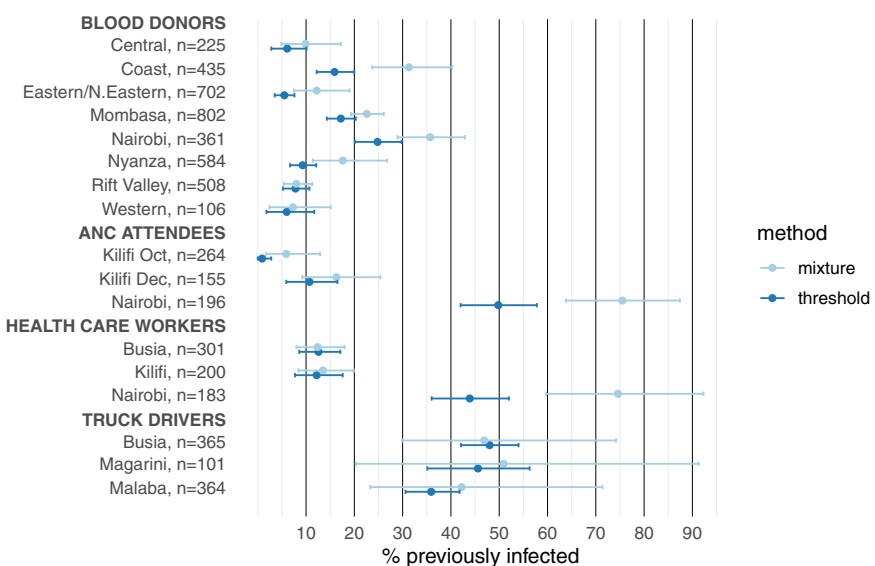

**Fig. 2 Mixture distributions fitted to anti-spike IgG antibody data collected in serological surveys of Kenyan blood donors, antenatal care (ANC) attendees, healthcare workers (HCW), and truck drivers.** The red distribution represents predicted responses in individuals previously infected with SARS-CoV-2 and the blue distribution represents predicted responses in previously uninfected individuals.

**Fig. 3 Previous SARS-CoV-2 infection in Kenyan blood donors, antenatal care attendees, health care workers and truck drivers estimated via threhold analysis and mixture modelling.** Estimates of the proportion previously infected are shown with 95% credible intervals.

the Coast region to estimate the variance, but then used this estimate to analyse data from other regions and from other subgroups such as pregnant women. Another limitation of the mixture modelling approach is that it is necessary to assume models for the component distributions. In our analysis, we assumed log antibody concentrations follow a normal distribution in previously uninfected individuals and a skew normal in previously infected individuals. However, we believe these distributional assumptions are not a major concern. The data from pre-COVID-19 samples suggests that log antibody levels in individuals who have not been infected are approximately normally distributed. And although we were unable to determine the distribution in previously infected individuals, the sensitivity analysis involving an alternative model and the analysis of simulated data both suggest that our estimates are robust to misspecification of this distribution. Finally, we note that our analysis was limited to Kenyan serosurveys; in the future it will be important to explore using mixture models to analyse surveys that have been done elsewhere.

We have shown that the threshold analysis produces estimates of the proportion previously infected that are likely to be biased downwards. While overestimating this proportion can lead to complacency in the assessment of future COVID-19 waves, underestimating it can also have serious adverse consequences if it prolongs social restrictions unnecessarily. This is particularly relevant in low-and-middle-income countries where resumption of educational, social and economic activities is unlikely to be brought about by rapid dissemination of COVID-19 vaccines. Here we provide an alternative to the standard threshold analysis that does not require specific adjustments for waning and allows for differences between populations in background antibody levels. The approach makes assumptions about variation in background antibody levels that need to be validated locally, but until we have a better understanding of the spectrum of antibody concentrations by symptom severity, or due to waning, it will probably be more accurate than the standard threshold analysis.

## Methods

**Data sources**. The blood samples were collected in studies of Kenyan blood donors[4,25], healthcare workers[26], truck drivers/assistants[27] and pregnant women[28]. Most surveys were done shortly before or during the early stages of the second wave of the epidemic in Kenya (Supplementary Fig. 4). The protocols for these studies were approved by the Scientific and Ethics Review Unit (SERU) of the Kenya Medical Research Institute. The blood donors and health care workers provided written informed consent, and the truck drivers provided verbal consent. Surveillance of antenatal care attendees was conducted at the request of the Kenya Ministry of Health and consent was obtained from participating health facilities and the respective Counties. The surveillance involved analysis of anonymised residual blood volumes of samples collected in antenatal care clinics. Approval to publish the results of the antenatal care surveillance was explicitly requested from and granted by Kenyatta National Hospital, University of Nairobi Ethics Review Committee (Protocol P327/06/2020) and the Kilifi County health management rapid response team and SERU.

**Enzyme-linked immunosorbent assay (ELISA)**. Across all serosurveys, the samples were tested for anti-SARS-CoV-2 IgG antibodies using an adaptation of the Krammer ELISA for whole length spike antigen[29]. Ratios of optical densities (OD) relative to a negative control were used to quantify the antibody concentrations. The assay was originally validated using 910 pre-COVID-19 serum samples collected in 2018, all of which were collected from adults and children from the Coast region of the country, and samples from 174 PCR-positive Kenyan adults, which were collected from patients admitted to Kenyatta National Hospital in Nairobi and their contacts (14 pre-symptomatic, 55 symptomatic, 92 asymptomatic and 13 unknown). For the samples obtained from PCR-positive individuals, the median time between the PCR test and blood sample collection was 21 days (IQR: 15, 34). The validation was based on a threshold OD ratio of 2, and yielded sensitivity and specificity estimates of 92.7% and 99.0% respectively. In a WHO-sponsored international standardisation study, the performance of the assay was found to be consistent with that of other assays[30].

**Statistical analysis**. Both the threshold-based analysis and the mixture model analysis were done using the Rstan package (version 2.21.2) in R version 4.0.4[31,32].

**Sensitivity and specificity adjusted threshold analysis**. We incorporated information on the sensitivity and specificity of the threshold by simultaneously modelling the serosurvey data and validation data. Specifically, we modelled counts of (i) the number, $y$, of survey samples above the threshold OD ratio, (ii) the number, $x$, of PCR-positive samples above the threshold and (iii) the number, $z$, of pre-COVID-19 samples below the threshold. In the model, the observed proportion of survey samples above the threshold, $p_{obs}$, is a function of the proportion previously infected, $p$, and the sensitivity and specificity of the threshold.

Model:

$$y \sim \text{Binomial}(p_{obs}, N)$$

$$x \sim \text{Binomial}(\text{se}, N_{se})$$

$$z \sim \text{Binomial}(\text{sp}, N_{sp})$$

$$p_{obs} = \text{se} \times p + (1 - \text{sp}) \times (1 - p).$$

Priors:

$$p \sim \text{Uniform}(0, 1)$$

$$\text{se} \sim \text{Uniform}(0, 1)$$

$$\text{sp} \sim \text{Uniform}(0, 1).$$

**Mixture model**. We fitted a two-component mixture model where individuals are classified according to whether or not they have experienced SARS-CoV-2 infection. We assumed that $\log_2$ OD ratios follow a skew normal among previously infected individuals (parameters: location = $\xi$, scale = $\omega$ and skew = $\alpha$) and a normal distribution among previously uninfected individuals (parameters: mean = $\theta$, standard deviation = $\nu$).

Model:

$$(1 - p) \times \text{Normal}(\theta, \nu) + p \times \text{Skew Normal}(\xi, \omega, \alpha),$$

where $p$ = proportion previously infected.

To make it easier to interpret the model parameters, we reparameterised the model in terms of the difference, $\delta$, between the means of the two distributions:

$$\xi = \theta + \delta - \omega \sqrt{2/\pi} \frac{\alpha}{\sqrt{1 + \alpha^2}}.$$

Since mixture models can be difficult to fit when there is overlap of the component distributions, we used several constraints to facilitate parameter estimation. First, we fixed the standard deviation in the uninfected, $\nu$, to be equal to the standard deviation in the pre-COVID-19 samples. Second, we used an informative prior for $\delta$ to constrain the difference in means—the prior puts 5% probability on the difference exceeding the difference between the mean in symptomatic PCR-positive cases (mean $\log_2$ OD ratios = 3.43) and the mean in pre-COVID-19 samples (mean $\log_2$ OD ratios = −0.17). The prior for $\delta$ was also designed to avoid label switching by ensuring $\delta > 0$. Finally, we used an informative prior for $\alpha$ to rule out strong skew in either direction.

Priors:

$$\theta \sim \text{Normal}(0, 10)$$

$$\delta \sim \text{Normal}^+(0, 1.83)$$

$$\ln \omega \sim \text{Normal}(0, 10)$$

$$\alpha \sim \text{Normal}(0, 1)$$

$$p \sim \text{Uniform}(0, 1).$$

In addition to estimating the model parameters, we estimated the sensitivity and specificity at various threshold values. The sensitivity corresponds to the proportion above the threshold in the skew normal distribution and the specificity corresponds to the proportion below the threshold in the normal distribution. Both quantities were estimated using the sample of parameter values drawn from the posterior distribution.

**An alternative specification of the mixture model**. As a sensitivity analysis, we fitted an alternative model where we assumed that the distribution among previously infected individuals follows a mixture distribution with mixing parameter $q$.

Model:

$$(1 - p) \times \text{Normal}(\theta_1, \nu_1) + p[q \times \text{Normal}(\theta_2, \nu_2) + (1 - q) \times \text{Normal}(\theta_3, \nu_3)].$$

As with the skew-normal model, the model was reparameterised in terms of the difference in mean between the positive and negative component, that is we defined $\theta_1 = q\theta_2 + (1 - q)\theta_3 - \delta$, where $\delta$ represents the difference between the means.

The priors for $\nu_i$ ($i = 2, 3$) were chosen to ensure that these standard deviations are of similar magnitude to the standard deviation observed in PCR-positive individuals (SD $\log_2$ OD ratios = 1.32 = $\exp(0.28)$) and we used the constraint $\theta_3 > \theta_2$ to avoid the problem of label switching and ensure the identifiability of these parameters.

Priors:

$$\delta \sim \text{Normal}^+(0, 1.83)$$

$$\theta_i \sim \text{Normal}(0, 10) \quad i = 2, 3 \quad \theta_3 > \theta_2$$

$$\ln \nu_i \sim \text{Normal}(0.28, 0.2) \quad i = 2, 3$$

$$p \sim \text{Uniform}(0, 1)$$

$$q \sim \text{Uniform}(0, 1).$$

**Reporting summary**. Further information on research design is available in the Nature Research Reporting Summary linked to this article.

## Data availability
Source data are provided in the supplementary materials (xlsx file). Source data are provided with this paper.

## Code availability
Stan code for fitting the Bayesian models is provided in Supplementary Notes 1–3 and R/ Stan code for all analyses, including the generation of tables and figures, is available at: https://github.com/christian-bottomley/mixture_model_sarscov2.

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

## Acknowledgements
We thank the serosurvey participants for their contribution to this research. This project was funded by the Wellcome Trust (grant numbers 220991/Z/20/Z, 203077/Z/16/Z), the Bill and Melinda Gates Foundation (INV-017547) and by the Foreign Commonwealth and Development Office (FCDO) through the East Africa Research Fund (EARF/ITT/ 039). J. Anthony G. Scott is funded by a Wellcome Trust Senior Research Fellowship (214320). Christian Bottomley is funded through an award that is jointly funded by the UK Medical Research Council (MRC) and the UK Foreign, Commonwealth and Development Office (FCDO) under the MRC/FCDO Concordat agreement and is also part of the EDCTP2 programme supported by the European Union (MR/R010161/1). For the purpose of open access, the authors have applied a CC BY public copyright licence to any Author Accepted Manuscript version arising from this submission. This work is published with the permission of the Director of the Kenya Medical Research Insitute.

## Author contributions
Conceptualisation: A.S. and C.B. Methodology: C.B., M.O. and A.S. Investigation: S.U., K.G., E.K., A.E., D.M., J.G., H.K., J.N., I.A., A.A. and G.W. Writing—original draft: C.B. Writing—review & editing: C.B., M.O., K.G., E.K., A.E., I.A., A.A., D.N. and A.S.

## Competing interests
The authors declare no competing interests.
