## [Peer Review File · Nature Communications]

REVIEWER COMMENTS

Reviewer #1 (Remarks to the Author):

The authors employ mixture modelling to estimate cumulative incidence of COVID-19. In comparison to threshold analysis, they show that the mixture model has a higher estimate. They do not comment on the validity of the results, if the cumulative infection result is reasonable.

The work presented here will be of interest to others as many are working to determine the true number of infections in their jurisdictions. The method could be adapted to other jurisdictions. However, comments on the realism of the actual estimates made here would benefit the study.

Reviewer #2 (Remarks to the Author):

Thank you for the opportunity to review this interesting manuscript, in which the authors use mixture models as an alternative to estimate cumulative infection incidence from seroprevalence data. They conclude that such methods result in higher estimates of infection incidence compared to threshold methods adjusted for test sensitivity and specificity. Conceptually, the approach is appealing and the methodology seems reasonable. I have some concerns about the results, which for some of the datasets used do not seem plausible. To me this indicates a need to provide some more stringent constraints to the seropositive distribution.

Major comments:

1. Could the authors provide more information on when these surveys were done relative to the timeline of the epidemic in each country? The authors use cross-sectional data, which represent a mix of recent and past infections. If the authors believe that waning is an important characteristic of neutralising antibody responses, as has been shown in a number of studies, then timing of sample collection relative to the epidemic peak is likely to be important in determining the distribution of antibody titres. Some comment on this in the discussion would also be useful to aid interpretation.

2. For the reference sample of 174 PCR positive patients, could the authors also provide information on when the serum samples were collected relative to the time of symptom onset, or earliest known time of infection if asymptomatic? Ideally one would use serum samples collected close to the timing of peak antibody levels, 3-4 weeks following infection, but it's not clear if this is the case here. From Figure 1, quite a few PCR positive individuals have negative or very low ELISA values, which I think is likely to influence their modelling results.

3. My main concern is that in Figures 2 and 3, the mixture models seem to yield results that seem highly implausible. For example, in Kenyan blood donors, the mixture model estimates a cumulative seroprevalence that is 4 times higher than the threshold model. For a number of other populations, the mixture model results in estimates that are some 50% higher than the threshold model. This doesn't seem plausible. In addition, some of the estimates, while being slightly higher than the threshold method, have much wider uncertainty bounds, indicating that this approach would be of limited practical use. I can't really think of a good reason why the assay would perform so differently in different populations, particularly when the assay validation showed very high sensitivity and specificity. Even accounting for the fact that serological assays tend to be validated with sera from more severe patients who likely have higher titres, such discrepancies don't seem reasonable. Additionally, looking at the mixture distributions in figure 2, for some populations, these allow for

seropositive individuals to have implausibly low titres. This seems to occur more often in populations with higher seroprevalence, which probably contributes to the discrepancy. I can think of a couple of reasons why this might be, e.g. the seropositive distribution is not sufficiently well specified and needs to be constrained in some way, or seropositives themselves represent a mixture of distributions (or both).

3. In the discussion, the authors discuss two explanations for the lower distribution of titres in seroprevalence data compared with PCR positives: lower peak responses in mild cases and asymptotically infected individuals, and waning of antibody titres over time. They suggest, citing other studies, that waning is a more likely explanation. But these two explanations aren't mutually exclusive. We know from longitudinal studies that asymptomatic individuals show broad heterogeneity in seroconversion patterns (see, e.g., this study in asymptotically infected migrant workers <https://rupress.org/jem/article/218/5/e20202617/211835/Highly-functional-virus-specific-cellular-immune>). To complicate things further, there is also wide heterogeneity in patterns of waning (see, e.g. <https://www.ncbi.nlm.nih.gov/pmc/articles/PMC7987301/>). To me this suggests that treating seropositives as a single distribution is probably not reasonable and probably results in distributions with very wide variance that contributes to the unrealistically high estimates in some of these populations.

Reviewer #3 (Remarks to the Author):

Review of NCOMMS-21-15873-T:

Improving SARS-CoV-2 cumulative incidence estimation through mixture modelling of antibody levels

Required information

1. What are the noteworthy results?

This study shows interesting data on the spread of SARS-CoV-2 infection in Kenya, in particular differences among different subpopulations. Interpretation of the data is studied by comparing a "traditional" cutoff-based method with a method employing binary distribution mixtures.

2. Will the work be of significance to the field and related fields? How does it compare to the established literature? If the work is not original, please provide relevant references.

The results will be interesting to sero-epidemiologists, the methods are not novel but because such methods are not widely used in serology it is certainly worthwhile to publish this application.

3. Does the work support the conclusions and claims, or is additional evidence needed?

The main claim - that the used threshold-based method tends to underestimate the prevalence - is valid. Some methodological details may require attention (details see below), and this reviewer suspects that the method may be simplified somewhat (I doubt the skew lognormal

distribution is needed).

4. Are there any flaws in the data analysis, interpretation and conclusions? - Do these prohibit publication or require revision?

The use of cumulative incidence is problematic. The term is ill-defined here, and I would prefer prevalence: that is simpler and correct. For instance, application of these results to steer distancing measures (as suggested in the discussion) is overreach. See below for specific comments.

The emphasis on MC methods may have obscured less glamorous analyses, like likelihood ratio methods for comparing (sub)populations and model variants.

5. Is the methodology sound? Does the work meet the expected standards in your field?

I see no problems, but having estimated mixture distributions, it would not have been hard to discuss these in terms of sensitivity/specificity. That could provide better insight into the meaning of the results, and appeal to serologists.

6. Is there enough detail provided in the methods for the work to be reproduced?

Yes.

General comments

This reviewer is not satisfied with the term cumulative incidence. Its definition (if that is what it is) in the introduction omits the period over which this measure is calculated. In the context of binary mixtures the fraction positive should be identified as prevalence. One might consider that SARS-CoV-2 has not been around very long, but the duration of the pandemic has now been long enough for considerable uncertainty over the actual incidence. Also, we still do not know much about the decay of these antibodies but it appears to be in the range of months (to possibly a few years), and experience with serology for a wide variety of infectious diseases indicates that magnitude and duration of the seroresponse to infection show considerable individual variation. Therefore, whenever a sample is (judged to be) positive, the time since infection may be anything from a few weeks to a year or more (for COVID). In fact, it has been shown that the higher the antibody level is, the more information this carries about the person sampled having been infected (recently).

Any checks for how sensitive the mixture model results are against misspecification of the shape of the component distributions?

Given the limited time that the COVID pandemic has been around, it

could be useful to try and exploit the sampling dates for the various surveys. Could some of the differences be explained by the likely times since introduction of SARS-CoV-2 in the region where these people resided? Samples collected soon after introduction are expected to have high antibody levels (as seen in the PCR positive samples) while persons sampled later, after the infection has been spreading for a while, would have a mix of times since infection, resulting in a large variance in antibody levels.

In a binary mixture model for antibody levels in a cross-sectional population sample, the two component distribution mixture may be interpreted as defining the probabilities for any observed antibody level to be assigned to the positive or negative component. These probabilities can be conveniently shown in a ROC diagram of sensitivity vs (1-) specificity. This has the advantage that failure to discriminate (between positives and negatives) are well visualized.

Specific comments

Introduction

p3-l6: A kind of implicit definition of cumulative incidence: the proportion that have been infected. That should be the prevalence. Those who have been infected during a specified period could be called the cumulative incidence over that period.

p3-l8: It might be useful to explain how the under-ascertainment may be mainly caused by the short period of positive PCR swabs, following infection, while humoral antibodies last a longer time. As such the PCR test is a quite reliable means of establishing current infection, just not of past infection.

p3-l10-14: I would have expected that the cutoff level was determined (how is it "estimated"?) using (samples from) PCR-positive cases, assuming some arbitrarily chosen sensitivity level, more in line with detection theory?

p3-l16-18: When trained on a sample not representative of the general population, spectrum bias is still possible.

p4-l13: Why fix the SD of the negative component? Just adding the pre-COVID samples as a a priori negative population could increase the information about the negative SD (and perhaps provide an opportunity for a check on whether that pre-COVID sample differs from the negative component in the COVID sample.

Results

p4-l21: significantly? I trust this was intended as a non-statistical observation (wasn't this a Bayesian analysis?)

p5-l15-6: When the skew parameter (allowing for extra-lognormal variance?) was close to zero, would it be worthwhile to check whether using a simple lognormal mixture would perform equally well?

p6-l1-2: How was model fit determined? What are considered plausible estimates?

Discussion

p7-l14-18: Why not use the mixture model to define better threshold levels? (I am playing devil's advocate here) But seriously: an adjusted cutoff may correct the underestimated prevalences for the threshold method.

p10 first paragraph: please note that here you are assuming that the logarithm of antibody level is normally distributed, not the antibody levels.

p10 second paragraph: The positive component of the binary mixture may have increased variance and thus account for the fact that subjects are sampled at varying times in their seroresponses. However, the fraction positives represents this same range of times since seroconversion, which, as we know by now, may span several months or even longer. Therefore, I would not think this fraction positives is a good variable to monitor the need for social distancing.

Figure 1: Interestingly, there seems to be a small subset of samples from PCR positive cases who have an OD ratio well within the range of the pre-COVID distribution. Is it known how soon after symptom onset these samples were taken? Could they be pre-seroconversion?

Reviewed by
PFM Teunis PhD
Center for Global Safe WASH, Rollins School of Public Health
Emory University, Atlanta, USA.

Response to reviewers' comments

We would like to thank the reviewers for their helpful comments. A detailed response to each of the comments is provided below.

Reviewer 1

The work presented here will be of interest to others as many are working to determine the true number of infections in their jurisdictions. The method could be adapted to other jurisdictions. However, comments on the realism of the actual estimates made here would benefit the study.

In the paper we argue that the standard threshold-based analysis of serological data is vulnerable to spectrum bias. The bias arises because recent/symptomatic infections are over-represented among the PCR-positive samples used to estimate sensitivity. Our hypothesis is consistent with Kenyan serosurvey data which show that antibody levels among survey participants do not segregate as clearly into positive and negative distributions as one would expect based on the distributions observed in pre-COVID-19 samples and PCR-positive samples.

Mixture models offer an alternative that does not rely on external estimates of sensitivity and specificity and is therefore potentially less vulnerable to bias.

To further illustrate the logic of our argument, we now present model-derived estimates of sensitivity (Supplementary Table 1). The model-derived estimates are consistently lower than sensitivity estimated from PCR-confirmed cases (average model-predicted sensitivity = 65% versus 93% sensitivity estimated using samples from PCR-confirmed cases) which "explains" why the threshold analysis estimates lower levels of previous infection than the mixture model analysis.

A weakness of the mixture modelling approach is that it is necessary to assume a particular form for the distribution of antibody levels among previously infected and uninfected individuals. In the revised paper we have fitted an alternative model where we assume a 2-component mixture in previously infected individuals (for the negative component we assume a single normal distribution as in the original model). The model produces similar results to the original model which lends additional support to our findings. The model is described on p.14/15 and the findings are discussed on p.5 and shown in Supplementary Figures 3 and 4.

Ideally, we would triangulate our results with other sources of data. Unfortunately, there are few sources of external data available, nevertheless our estimates are broadly consistent with recent modelling by Brand et al. (cited in the paper) which suggests that 50% of the Kenyan population had been infected by the end of 2020.

In the long run we will be able to test the mixture modelling approach as further serological data become available. For example, we will be able to compare the mixture model and threshold analyses in surveys where two or more assays have been used. Our expectation is that the mixture modelling will produce estimates that are consistent across different assays, whereas the threshold approach will produce estimates that vary depending on the degree of antibody waning and ability of the assay to detect asymptomatic infection.

Reviewer 2

1. Could the authors provide more information on when these surveys were done relative to the timeline of the epidemic in each country? The authors use cross-sectional data, which represent a mix of recent and past infections. If the authors believe that waning is an important characteristic of neutralising antibody responses, as has been shown in a number of studies, then timing of sample collection relative to the epidemic peak is likely to be important in determining the distribution of antibody titres. Some comment on this in the discussion would also be useful to aid interpretation.

The first wave of COVID-19 in Kenya started in March 2020 and continued until the middle of August. There was then a period of limited transmission before the second wave which started in the middle of October and continued until the end of December. Since the serosurveys were mainly done in the period August – November, i.e., after the peak of the first wave, we expect the distribution of antibodies to reflect a mixture of recent and historic infections. The revised manuscript includes a figure that shows the total number of cases in Kenya as well the timing of the various surveys (Supplementary Figure 1). We thank the reviewer for this suggestion.

2. For the reference sample of 174 PCR positive patients, could the authors also provide information on when the serum samples were collected relative to the time of symptom onset, or earliest known time of infection if asymptomatic? Ideally one would use serum samples collected close to the timing of peak antibody levels, 3-4 weeks following infection, but it's not clear if this is the case here. From Figure 1, quite a few PCR positive individuals have negative or very low ELISA values, which I think is likely to influence their modelling results.

The median time between the PCR test and blood sample was 21 days (IQR: 15, 34). The median was previously presented in the Discussion, but we have moved this background information to the Methods (p.11).

The mixture model does not use data on the PCR positives and is therefore not influenced by the distribution in this sample. We argue that this is a strength of the mixture modelling approach because it does not rely on the assumption that the PCR-confirmed positives are representative of positives in the wider population.

It is difficult to know why some PCR-positive cases had low antibody levels. Possible explanations include that the PCR-result was incorrect (false positives) or that individuals were sampled before they had the opportunity to produce antibodies. Alternatively, the low levels could reflect natural variation. These possibilities are discussed in further detail below in our response to a similar question raised by review 3.

3. My main concern is that in Figures 2 and 3, the mixture models seem to yield results that seem highly implausible. For example, in Kenyan blood donors, the mixture model estimates a cumulative seroprevalence that is 4 times higher than the threshold model. For a number of other populations, the mixture model results in estimates that are some 50% higher than the threshold model. This doesn't seem plausible. In addition, some of the estimates, while being slightly higher than the threshold method, have much wider uncertainty bounds, indicating that this approach would be of limited practical use. I can't really think of a good reason why the assay would perform so differently in different populations, particularly when the assay validation showed very high sensitivity and specificity. Even accounting for the fact that serological assays tend to be validated with sera from more severe patients who likely have higher titres, such discrepancies don't seem reasonable.

Additionally, looking at the mixture distributions in figure 2, for some populations, these allow for seropositive individuals to have implausibly low titres. This seems to occur more often in populations with higher seroprevalence, which probably contributes to the discrepancy. I can think of a couple of reasons why this might be, e.g. the seropositive distribution is not

sufficiently well specified and needs to be constrained in some way, or seropositives themselves represent a mixture of distributions (or both).

The mixture model estimates are either close to or greater than the threshold estimates. The problem they aim to solve is a known issue – spectrum bias. The corrections applied to the raw prevalence are based on measured sensitivity and we know that the sensitivity of any threshold is highest at 3-6 weeks after infection, and higher in those with severe symptoms – precisely the sample set used to estimate sensitivity. If we were to estimate sensitivity in asymptomatic individuals at 6 months, it would be lower and therefore the adjustment to the raw prevalence would be higher – much as predicted by the mixture model.

We agree with the reviewer that fitting the mixture model to the combined Kenyan data produces an estimate that is implausibly high. In the Results and Discussion, we argue that this is because the assumed background variation does not account for between-region variation and is therefore an underestimate. On reflection, we think that presenting this analysis as part of the main analysis is confusing and have therefore removed it from Figures 2 & 3. We still mention in the Results and Discussion the problem of combining data from different serosurveys, and we use the Kenya-wide analysis to illustrate the problem.

With the exception of the Kenya-wide estimate, we believe the mixture model estimates are plausible. The absence of a clear separation between positives and negatives in the serosurvey data—as one would expect based on the distributions in pre-COVID-19 and PCR positive samples shown in Figure 1—strongly suggests that sensitivity is overestimated and that the true proportion positive is higher than predicted by the standard analysis. Consistent with this, the mixture model produces estimates that are higher than the threshold analysis across almost all of the analyses.

The fact that there is variation in the extent to which the threshold method underestimates the proportion previously infected is to be expected. Some of this variation is due to chance (the confidence intervals are often wide) but there is also likely to be variation between populations in the fraction of asymptomatic cases and the degree of antibody waning – both of these factors introduce bias into the standard threshold estimate.

We agree that it is not ideal to have wide confidence intervals but in our view it is preferable to relying on a sensitivity estimate that is likely to be biased, i.e. it is better to sacrifice some precision to avoid bias. We note that wide confidence intervals generally occur when there is significant overlap of the component distributions (intuitively this is because it is difficult in this scenario to disentangle the component distributions). When there is less overlap, the mixture model analysis produces confidence intervals that are only slightly wider than the threshold analysis.

The reviewer correctly identifies the distributional assumptions as a concern. The assumption that antibody levels in uninfected individuals follow a normal distribution is corroborated by the distribution in pre-COVID samples (Figure 1). It is more difficult to validate the assumption of a skew-normal distribution in individuals who have been infected. Previously we tested this assumption by fitting the model to data simulated from alternative distributions. In the revised manuscript, we now also present a sensitivity analysis where instead of assuming a skew-normal we assume a mixture of normals for the positive component (i.e. the negative component is a single normal distribution and the positive component is a mixture of two normal distributions). The model is described on p.14/15. The estimates from this analysis (Supplementary Figures 3 and 4) are similar to those from the original analysis. We are therefore reasonably confident that the methodology is robust to the distributional assumptions that we have made.

4. In the discussion, the authors discuss two explanations for the lower distribution of titres in

seroprevalence data compared with PCR positives: lower peak responses in mild cases and asymptomatically infected individuals, and waning of antibody titres over time. They suggest, citing other studies, that waning is a more likely explanation. But these two explanations aren't mutually exclusive. We know from longitudinal studies that asymptomatic individuals show broad heterogeneity in seroconversion patterns (see, e.g., this study in asymptomatically infected migrant workers <https://rupress.org/jem/article/218/5/e20202617/211835/Highly-functional-virus-specific-cellular-immune>).

To complicate things further, there is also wide heterogeneity in patterns of waning (see, e.g. <https://www.ncbi.nlm.nih.gov/pmc/articles/PMC7987301/>). To me this suggests that treating seropositives as a single distribution is probably not reasonable and probably results in distributions with very wide variance that contributes to the unrealistically high estimates in some of these populations.

We agree that both waning and no. of symptoms are likely to be important determinants of antibody levels. We have rewritten the paragraph (p.7) to give equal weight to both explanations and have included the paper by Chia in the Discussion (we thank the reviewer for drawing our attention to this paper).

As mentioned in our response to the reviewer's previous comment, we have now conducted a sensitivity analysis where we assume a mixture distribution in previously infected individuals. This analysis produces similar estimates (Supplementary Figure 4). We are therefore reasonably confident that the model adequately accounts for variability between different subgroups within the positives.

Reviewer 3

General comments

This reviewer is not satisfied with the term cumulative incidence. Its definition (if that is what it is) in the introduction omits the period over which this measure is calculated. In the context of binary mixtures the fraction positive should be identified as prevalence. One might consider that SARS-CoV-2 has not been around very long, but the duration of the pandemic has now been long enough for considerable uncertainty over the actual incidence. Also, we still do not know much about the decay of these antibodies but it appears to be in the range of months (to possibly a few years), and experience with serology for a wide variety of infectious diseases indicates that magnitude and duration of the seroresponse to infection show considerable individual variation. Therefore, whenever a sample is (judged to be) positive, the time since infection may be anything from a few weeks to a year or more (for COVID). In fact, it has been shown that the higher the antibody level is, the more information this carries about the person sampled having been infected (recently).

Following Bouman et al. (2021) we previously used "cumulative incidence" to mean the proportion of the population infected between the start of the epidemic and the time of the serosurvey. However, we appreciate that because the start of the epidemic is not known exactly this terminology might not be universally acceptable. We have therefore changed the terminology as suggested, and now refer to the estimates from the mixture model analysis (and threshold analysis) as estimates of the "proportion previously infected".

Any checks for how sensitive the mixture model results are against misspecification of the shape of the component distributions?

In the original manuscript we tested the mixture model on simulated data. In the revised manuscript, we now also show that the mixture modelling produces similar results under an

alternative specification in which log antibody levels in previously infected individuals follow a 2-component mixture model. Because the results of the two analyses are consistent, we are reasonably confident that the mixture modelling approach is robust to the distributional assumptions that we have made. The model used in this sensitivity analysis is described on p.14/15 and results are shown in Supplementary Figures 3 and 4.

Given the limited time that the COVID pandemic has been around, it could be useful to try and exploit the sampling dates for the various surveys. Could some of the differences be explained by the likely times since introduction of SARS-CoV-2 in the region where these people resided? Samples collected soon after introduction are expected to have high antibody levels (as seen in the PCR positive samples) while persons sampled later, after the infection has been spreading for a while, would have a mix of times since infection, resulting in a large variance in antibody levels.

This is an interesting suggestion and an avenue that might be worth exploring in the future. We have not attempted to incorporate time since infection in the present analysis for two reasons. First there is little variation in the timing of the surveys presented in the current analysis (Supplementary Figure 1). Second there is limited PCR testing in Kenya, with most of the testing being done in Nairobi and to a lesser extent Mombasa, so it is difficult to know when SARS-CoV-2 was introduced in each region.

In addition, incorporating time since infection would complicate the mixture model and there is a risk that the main message of the paper—that the standard approach tends to underestimate infection—would be lost as a result.

In a binary mixture model for antibody levels in a cross-sectional population sample, the two component distribution mixture may be interpreted as defining the probabilities for any observed antibody level to be assigned to the positive or negative component. These probabilities can be conveniently shown in a ROC diagram of sensitivity vs (1-) specificity. This has the advantage that failure to discriminate (between positives and negatives) are well visualized.

ROC curves have been included in the revised manuscript (Supplementary Figure 2). We have also added model-based estimates of sensitivity and specificity (for the threshold OD ratio >2) to the table of parameter estimates (Supplementary Table 1). We thank the reviewer for this excellent suggestion.

Specific comments

Introduction

p3-l6: A kind of implicit definition of cumulative incidence: the proportion that have been infected. That should be the prevalence. Those who have been infected during a specified period could be called the cumulative incidence over that period.

We now use “proportion previously infected” instead of “cumulative incidence” throughout the paper.

p3-l8: It might be useful to explain how the under-ascertainment may be mainly caused by the short period of positive PCR swabs, following infection, while humoral antibodies last a longer time. As such the PCR test is a quite reliable means of establishing current infection, just not of past infection.

To simplify, we have removed the discussion of PCR testing. The sentence (p.3) is now:

“The proportion previously infected is usually established through serological surveys (i.e. data on antibody levels).”

p3-l10-14: I would have expected that the cutoff level was determined (how is it "estimated"?) using (samples from) PCR-positive cases, assuming some arbitrarily chosen sensitivity level, more in line with detection theory?

We have adopted the reverse perspective here—first the threshold is chosen (2 or 3 SDs above the mean in the known negative population is a common choice) and then the sensitivity/specificity is estimated.

p3-l16-18: When trained on a sample not representative of the general population, spectrum bias is still possible.

We have added a sentence (p.3) to clarify this point:

“Bias of this kind, that is bias that arises because sensitivity (or specificity) is estimated in a non-representative sample, is often referred to as spectrum bias.”

p4-l13: Why fix the SD of the negative component? Just adding the pre-COVID samples as a priori negative population could increase the information about the negative SD (and perhaps provide an opportunity for a check on whether that pre-COVID sample differs from the negative component in the COVID sample.

We agree with the reviewer that ideally we would estimate the SD in the negative component using an informative prior (i.e. using information from pre-COVID samples). However, in practice when we tried to do this convergence was often an issue, and we therefore chose to fix the parameter.

Results

p4-l21: significantly? I trust this was intended as a non-statistical observation (wasn't this a Bayesian analysis?)

Yes, this was a Bayesian analysis. We have deleted “significantly” in this sentence to avoid any ambiguity.

p5-l5-6: When the skew parameter (allowing for extra-lognormal variance?) was close to zero, would it be worthwhile to check whether using a simple lognormal mixture would perform equally well?

The simpler normal model produces similar results (as does the more complicated model where the distribution in previously infected individuals is a mixture). We have chosen the skew-normal model as our primary analysis because there is a hint of skew in some surveys (e.g. Truck drivers in Busia) and because it seems to be a good compromise between flexibility on the one hand and simplicity on the other.

p6-l1-2: How was model fit determined? What are considered plausible estimates?

Model fit was assessed visually. We agree that this sentence was vague and have deleted it from the revised version.

Discussion

p7-l14-18: Why not use the mixture model to define better threshold levels? (I am playing

devil's advocate here) But seriously: an adjusted cutoff may correct the underestimated prevalences for the threshold method.

We now present model-derived sensitivity and specificity estimates for all the surveys (Supplementary Table 1 and Supplementary Figure 2). As the reviewer suggests, these can be used to improve the threshold method. They also help to understand why the mixture model produces higher estimates of the proportion previously infected.

We thank the reviewer for this suggestion.

p10 first paragraph: please note that here you are assuming that the logarithm of antibody level is normally distributed, not the antibody levels.

We have corrected this error and checked the revised manuscript for similar errors.

p10 second paragraph: The positive component of the binary mixture may have increased variance and thus account for the fact that subjects are sampled at varying times in their seroresponses. However, the fraction positives represents this same range of times since seroconversion, which, as we know by now, may span several months or even longer. Therefore, I would not think this fraction positives is a good variable to monitor the need for social distancing.

There is now good evidence from longitudinal follow up of PCR-positive or seropositive (symptomatic and asymptomatic) individuals that previous infection strongly protects against reinfection (see for example cohort studies by Abu-Raddad et al. and Lumley et al. included in the references). Therefore, basing policy decisions on estimates of previous infection seems reasonable to us, provided that uncertainty in these estimates is taken into account. We think it is preferable to use estimates of the proportion previously infected rather than crude estimates of seropositivity for two reasons. Firstly, antibody-levels do not necessarily reflect the degree of T-cell immunity, which is known to be an important role in SARS-CoV-2 infection. Secondly, antibody waning depends on the assay used; therefore, depending on the assay, the measured antibody-levels may not adequately reflect the humoral response.

Figure 1: Interestingly, there seems to be a small subset of samples from PCR positive cases who have an OD ratio well within the range of the pre-COVID distribution. Is it known how soon after symptom onset these samples were taken? Could they be pre-seroconversion?

This is an interesting observation. The low responses in some PCR-positive cases do not seem to be associated with a short time interval between PCR testing and blood sample collection. Among 18 cases below the threshold, the time between PCR testing and blood testing was less than 14 days in only 5/18 of these cases.

The low response cases could reflect natural variation. In support of this hypothesis, we note that the standard deviation estimated for the positive component in many surveys was similar to the standard deviation in PCR-positive cases. Alternatively, it is conceivable that many of the low response cases are PCR false positives. For example, assuming 100% sensitivity, 99% specificity, and a 10% rate of positive tests (a typical rate of positivity), we would expect a 91% positive predictive value—so the idea that 18 (out of 174) PCR positives are false is not unreasonable.

REVIEWERS' COMMENTS

Reviewer #2 (Remarks to the Author):

Thank you for the opportunity to review the revised version of this manuscript. I appreciate the authors' detailed responses to my original comments and the additional analyses, particularly around the sensitivity of the model to different distributional assumptions. The revisions help to clarify most of the original concerns and I have no further comments. I do still have some reservations about the wide discrepancies and uncertainty bounds in higher prevalence settings compared with the threshold method, and how useful this is in practice, although I appreciate that some of this might reflect features of the local epidemiology or, as the authors suggest, geographic variations that are not disaggregated in the data.